# Inequalities in the uptake of, adherence to and effectiveness of behavioural weight management interventions: systematic review protocol

Jack Michael Birch  ,[1] Simon J Griffin,[1,2] Michael P Kelly,[2] Amy L Ahern  [1]

¹MRC Epidemiology Unit, University of Cambridge School of Clinical Medicine, Cambridge, UK
²Primary Care Unit, University of Cambridge School of Clinical Medicine, Cambridge, UK

**Correspondence to**
Jack Michael Birch;
jack.birch@mrc-epid.cam.ac.uk

## ABSTRACT

**Introduction** It has been suggested that interventions focusing on individual behaviour change, such as behavioural weight management interventions, may exacerbate health inequalities. These intervention-generated inequalities may occur at different stages, including intervention uptake, adherence and effectiveness. We will synthesise evidence on how different measures of inequality moderate the uptake, adherence and effectiveness of behavioural weight management interventions in adults.

**Methods and analysis** We will update a previous systematic literature review from the United States Preventive Services Taskforce to identify trials of behavioural weight management interventions in adults aged 18 years and over that were, or could feasibly be, conducted in or recruited from primary care. Medline, Cochrane database (CENTRAL) and PsycINFO will be searched. Only randomised controlled trials (RCTs) and cluster-RCTs will be included. Two investigators will independently screen articles for eligibility and conduct risk of bias assessment. We will curate publication families for eligible trials. The PROGRESS-Plus acronym (place of residence, race/ethnicity, occupation, gender, religion, education, socioeconomic status, social capital, plus other discriminating factors) will be used to consider a comprehensive range of health inequalities. Data on trial uptake, intervention adherence, weight change and PROGRESS-Plus-related data will be extracted. Data will be synthesised narratively. We will present a Harvest plot for each PROGRESS-Plus criterion and whether each trial found a negative, positive or no health inequality gradient. We will also identify potential sources of unpublished original research data on these factors which can be synthesised through a future individual participant data meta-analysis.

**Ethics and dissemination** Ethical approval is not required as no primary data are being collected. The completed systematic review will be disseminated in a peer-reviewed journal, at conferences, and contribute to the lead author's PhD thesis. Authors of trials included in the completed systematic review may be invited to collaborate on a future individual participant data meta-analysis.

**PROSPERO registration number** CRD42020173242.

## Strengths and limitations of this study

► A description of existing data that relate to inequalities in behavioural weight management trials, and where they occur will be provided, enabling future meta-analysis of individual participant data.
► A comprehensive search strategy, which has previously been validated in the literature, will be used to identify relevant trials.
► Where data permit, subgroup analysis of association or interaction between the PROGRESS-Plus criteria (place of residence, race/ethnicity, occupation, gender, religion, education, socioeconomic status, social capital, plus other discriminating factors) and trial uptake, adherence and effectiveness will be presented in a Harvest plot.
► Only randomised controlled trials (RCTs) and cluster RCTs will be included.
► There is likely heterogeneity in measures of PROGRESS-Plus criteria used, as well as limited publication of data, which is likely to prevent a meta-analysis being conducted.

## INTRODUCTION
### Rationale
Overweight and obesity are associated with an increased risk of a number of non-communicable diseases such as type 2 diabetes, cardiovascular disease and some cancers (including post-menopausal breast, bowel and oesophageal).[1 2] People living with overweight and obesity have greater all-cause mortality compared with those within a healthy weight range.[3] There are known health inequalities by place of residence, ethnicity, occupation, sex, religion, education, socioeconomic status (SES), social capital and other factors such as disability and sexual orientation (PROGRESS-Plus).[4] Observational research suggests that inequalities in overweight and obesity exist across several of these criteria, such as SES and education,[5–10] although these measures are generally more predictive of obesity in women than men.[9 11]

Both 'upstream' and 'downstream' interventions are needed to reduce the prevalence of obesity through primary prevention and treatment for those living with overweight and obesity. There is suggestion that 'upstream' interventions - that is, those aimed at a population-level and requiring little personal agency - are the most equitable,[12] and may reduce inequalities in overweight and obesity prevalence. On the other hand, 'downstream' interventions, targeted at high-risk groups and individuals (such as those who already have overweight or obesity) and requiring high personal agency, are likely to be inequitable. Inequitable interventions may exacerbate health inequalities if they are less effective at reducing overweight and obesity prevalence in disadvantaged groups. Behavioural weight management interventions, such as those provided in or referred to from primary care, require a high level of personal agency as participants are required to attend and be engaged with an intervention for it to be effective.[13] Hence, behavioural weight management interventions may inadvertently exacerbate health inequalities.

The overall effectiveness of behavioural weight management interventions was considered in a systematic review and meta-analysis for the United States Preventive Services Task Force (USPSTF).[14] The review considered behavioural weight loss and behavioural weight loss maintenance interventions, as well as pharmacological weight loss and weight loss maintenance interventions. It found that primary care-relevant behavioural weight loss interventions were associated with greater mean weight loss at 12–18 months when compared with a control, while behavioural weight loss maintenance interventions are effective at preventing weight regain. Moderation of effectiveness by any of the PROGRESS-Plus criteria was not considered, although narrative comment was made about the reporting of ethnicity and SES. Unless a specific ethnicity was targeted in the intervention, the authors found that ethnicity and SES were not well reported. Where ethnicity and SES were reported, most participants were white and of mid-to-high SES. Income, employment and/or occupation were the most frequently reported measures of SES.

We identified one previous systematic review from Hillier-Brown *et al* that considered the effectiveness of individual-level, community-level and societal-level interventions at reducing socioeconomic inequalities in obesity.[11] The individual (n=5) and community interventions (n=12) included in the systematic review were similar to the behavioural interventions included in the USPSTF review. They defined individual-level interventions as being conducted in a healthcare, research or home setting and delivered one-to-one. Community-level interventions were defined as being delivered to a group and taking place in community settings such as in community or sports centres. The review found that, for individual-level interventions, evidence for reducing inequalities in obesity among adults was only found in tailored weight loss programmes targeted at low-income

groups, particularly those in primary care settings, rather than for 'universal' interventions. Evidence was generally only for short-term outcomes (up to 9 months). Community-level interventions showed positive effects up to 3 months, although there was no evidence for longer-term positive effects. Meanwhile, there was little evidence for the impact of societal-level ('upstream') interventions on inequalities in obesity among adults, and the included evidence was of low quality.

There are some limitations of the Hillier-Brown *et al* review. A meta-analysis was not conducted, due to heterogeneity of the included studies. While a highly sensitive search strategy was used, only literature works that reported differential effects by a measure of SES were included. This meant that interventions which may have collected data on SES, but had not included it in analyses reported in a published paper, would have been excluded. The authors highlighted this may explain why mostly interventions taking a targeted approach to reducing SES inequalities were included; only a minority of studies examined intervention effects across the SES gradient. They also suggest that this targeted approach 'has limitations as even when interventions are effective among low-income groups they are only able to reduce the health inequalities gap, they have little effect on the wider social gradient'. Literature published since Hillier-Brown *et al* has considered the effect of universal interventions on health inequalities among adults.[15 16]

The Hillier-Brown *et al* review only considered inequalities in intervention effectiveness. Intervention-generated inequalities may occur at several stages.[12] First, in intervention uptake.[17] This may occur because of differing levels of weight loss service provision by geography or because some groups are less likely to take up the offer of an intervention. Research using the Clinical Practice Research Datalink in the UK found that certain groups, such as those in deprivation, may be more likely to access weight management interventions,[18] suggesting that such interventions may have a positive effect on health inequalities. Second, inequalities may occur in the adherence to an intervention.[19] Adherence to an intervention may be affected by certain barriers such as access to transport,[20] insufficient time or other social circumstances. Third, there may be inequalities in outcome—those of a certain socioeconomic position or ethnicity may have similar uptake and adherence to an intervention, but there may be other factors that mean that the intervention is less effective for them than for other people. This may be because the intervention is not culturally or contextually tailored appropriately.

In the current systematic review, we will synthesise literature on inequalities across the uptake, adherence and effectiveness of behavioural weight management interventions. The lack of reporting or analysis by measures associated with the PROGRESS-Plus criteria identified in both the USPSTF and Hillier Brown *et al's* systematic reviews suggest that it is not possible to fully explore inequalities in the effectiveness using aggregated data from published

literature alone. This lack of reporting may have occurred because individual trials may not be sufficiently powered to detect an interaction between moderators such as SES and the outcome; they are likely just to be sufficiently powered to detect the main, overall effect. This systematic review will also identify trials with unpublished data on measures of inequality across the PROGRESS-Plus criteria in order to conduct a future individual participant data meta-analysis.

## Objectives

The overall aim is to identify and describe inequalities in the update, adherence and effectiveness of behavioural weight management interventions. We will meet this aim through the following objectives: (1) to synthesise published literature on how inequalities across different PROGRESS-Plus criteria moderate the uptake, adherence and effectiveness of behavioural weight management interventions; and (2) to identify published trials that have unpublished data on how inequalities across different PROGRESS-Plus criteria moderate the uptake, adherence and effectiveness of behavioural weight management interventions.

## METHODS

This protocol was written in accordance with the Preferred Reporting Items for Systematic Review and Meta-Analysis Protocols (PRISMA-P) reporting guidelines for systematic review protocols (online supplemental file A).[21]

### Study design

We will conduct a systematic review of published randomised controlled trials (RCTs) of behavioural weight management interventions (which includes interventions for both behavioural weight loss and behavioural weight loss maintenance). PROGRESS-Plus criteria-related measures and data (outlined in table 1) will be extracted and evidence regarding their impact on uptake, adherence and effectiveness will be synthesised. Furthermore, we will identify where data relating to the PROGRESS-Plus criteria have been collected and their relationship with uptake, adherence and effectiveness not analysed, to facilitate future individual participant data meta-analysis.

Initially, relevant literature concerning behavioural weight management trials will be extracted from the 2018 USPSTF systematic review of interventions to prevent obesity-related morbidity and mortality in adults.[14] Then, a search of the same databases used in the USPSTF review will be conducted to identify trials published since the search was completed for the USPSTF systematic review on 6th of June 2017. We will use the same search strategies and terms as in the original report, but with pharmacological interventions excluded and terms relating to adverse events removed.

### Eligibility criteria

We will select studies according to the criteria outlined below.

### Study designs

We will include research articles reporting RCTs and cluster-RCTs. To mirror the USPSTF review, only studies published in the English language will be included.

### Participants

We will include studies of adults aged 18 years and over with overweight or obesity (body mass index >25 kg/m$^2$) who are suitable for behavioural weight loss or behavioural weight loss maintenance interventions. Participants may have additional risk factors such as hypertension, dyslipidaemia, impaired glucose tolerance or impaired fasting glucose.

Studies will be excluded if the population: was not selected based on a weight-related measure; had secondary causes of obesity (such as steroid use); selected on the basis of having a chronic disease for which behavioural weight loss or behavioural weight loss maintenance is part of disease management; was of pregnant women; was of adults in institutions or if the intervention was targeted at parents in order to change the behaviour of children.

### Interventions

Studies will be included if they were conducted in or recruited from primary care or a healthcare system, or could feasibly be implemented in or referred to from primary care. In the case of the latter, the interventions must be conducted as part of a healthcare setting or be available in the community at a national level, such as commercial weight loss interventions. We will include behavioural interventions that are focused on weight loss or weight loss maintenance. Interventions may be delivered either alone or as part of a multicomponent intervention on wider diet and nutrition, physical activity, sedentary behaviour or a combination of these. The intervention may include (but not limited to): assessment with feedback, advice, collaborative goal-setting, assistance, exercise prescriptions (referral to exercise facility or programme), arranging further contacts or provider training.

The delivery of the intervention may be: face-to-face contact, telephone, print materials, or be computer-based or mobile phone-based technology (such as websites, apps or text messages). There is no restriction on who delivers the intervention.

Interventions of alternative and complementary treatments (eg, mindfulness) will be excluded. All pharmacological and surgical interventions will be excluded, including in combination with behavioural interventions, unless the trial includes behavioural only and control arms.

### Comparators

We will only include trials with a control group. The control group may receive no intervention (wait-list

**Table 1** Definition of PROGRESS-Plus factors (adapted from Attwood *et al*)

| PROGRESS-Plus factor | Description | Example measures |
|---|---|---|
| Place of residence | Places, and perceptions of, where individuals live | ► Postcode<br>► Country, state, region, town or community<br>► Urban/rural<br>► Housing characteristics<br>► Distance to attend weight loss session<br>► Local food environment<br>► 'Walkability' |
| Race/ethnicity | Racial or ethnic group, or other classification of culture, language or nationality status | ► Ethnicity classifications<br>► Country of origin<br>► Language<br>► Other classifications of culture |
| Occupation | Occupational situation, patterns of work or features of working environment | ► Professional/skilled/unskilled/unemployed<br>► Unemployed/employed/retired<br>► Full time/part time<br>► Manual/non-manual |
| Gender/sex | Gender is self-identified by individuals, incorporating ideas around socially constructed roles and behaviours Sex refers to biological and physiological characteristics that define an individual as a man or woman | ► Gender<br>► Sex (eg, male/female classifications) |
| Religion | Religious affiliation or system of religious/spiritual beliefs or values | ► Religious denomination |
| Education | Extent and type of education or other formal training | ► Years in education<br>► Level of education attained (eg, for UK: GCSE, A-Levels, Undergraduate)<br>► Institutions attended (eg, for USA: high school/some college/college graduate/ university) |
| Socioeconomic status | An individual's position within a hierarchical social structure. Measures of socioeconomic status aim to capture access to resources, privilege, power or control | ► Indices of Multiple Deprivation (UK only, Scottish Index of Multiple Deprivation)<br>► Social class<br>► Individual income<br>► Household income<br>► Receiving state welfare (eg, benefits/free prescriptions in the UK, Medicaid in the USA)<br>► Asset-based measures (eg, home or car ownership)<br>► Occupation (eg, occupation class) |
| Social capital | Social capital aims to capture the obligations and benefits conferred on an individual by their society and social relationships. Can be viewed as a measure of interconnectedness between an individual and their social surroundings or group | ► Marital/relationship status (eg, single, cohabiting)<br>► Household size<br>► Social support<br>► Social networks<br>► Civic participation/group membership<br>► Ability to use technology |

Continued

| PROGRESS-Plus factor | Description | Example measures |
|---|---|---|
| Plus | Any other factors over an individual's life course that could lead to discrimination. Examples include age, disability and sexual orientation | ► Self-reported age in years<br>► Measures of health status and/or quality of life (eg, EuroQoL, SF-36, EQ-5D)<br>► Tests of physical function<br>► Physical or emotional/mental disability<br>► Self-reported sexual orientation (eg, heterosexual, homosexual, bisexual) |

GCSE, General Certificate of Secondary Education; SF-36, 36-Item Short Form.

control or usual care) or minimal intervention (such as generic print or electronic materials).

## Outcomes and prioritisation

Outcomes will occur at the three stages: uptake, adherence/attendance and effectiveness.

Differential uptake will be considered at two stages. First, trial uptake will be calculated for each study using the formula:

$$\frac{Participants\ accepting\ invitation\ to\ trial}{Participants\ invited\ to\ trial}$$

Second, uptake of the intervention arm. We are considering uptake of the intervention to be 'attending' at least one intervention session, the language of which is geared towards to those attending community group interventions such as WW (formerly Weight Watchers). 'Attendance' to an online-based intervention would be defined as logging into the online platform at least once. Hence, uptake of the intervention is defined as:

$$\frac{Participants\ attending\ at\ least\ one\ intervention\ session}{Participants\ in\ trial\ arm}$$

The second outcome stage is adherence to the intervention. We will consider adherence for each participant as either a binary variable (adhered vs not adhered) or using the below formula, depending on how adherence is defined in the included studies.

$$\frac{Number\ of\ sessions\ attended}{Number\ of\ sessions\ prescribed}$$

The final outcome stage is effectiveness. This will be assessed at the 12-month follow-up using three measures: weight change in kilograms, weight loss of 5% or greater, and change in waist circumference.

## Timing (eg, minimum follow-up)

As per the USPSTF review, we will only include studies that measure intervention effectiveness at 12 or 18 months. This is despite the different timings required for each of the outcomes. The uptake outcomes require data at two pre-intervention stages—invitation to trial and baseline—as well as data on percentage of participants attending at least one session of an intervention. The adherence outcome will require data from baseline until the end of the intervention. Finally, intervention effectiveness will be assessed at 12 months or later from baseline.

## Moderator variables

The moderator variables under consideration are the PROGRESS-Plus criteria. Possible measures of each of these criteria are shown in table 1, which has been adapted from a systematic review that explored equity in primary care-based physical activity interventions using PROGRESS-Plus.[22]

## Setting

Eligible studies will have been conducted in primary care, referred from primary care or be applicable to primary care settings. As per the search performed in the USPSTF review, only studies conducted in countries that were members of the Organisation for Economic Co-operation and Development (as of 2017) are eligible for inclusion. These countries are: Australia, Austria, Belgium, Canada, Chile, Czech Republic, Denmark, Estonia, Finland, France, Germany, Greece, Hungary, Iceland, Israel, Italy, Japan, South Korea, Luxembourg, Mexico, the Netherlands, New Zealand, Norway, Poland, Portugal, Slovakia, Slovenia, Spain, Sweden, Switzerland, Turkey, the UK and the USA.

## Information sources and search strategy
### Electronic searches

It is anticipated that much of the PROGRESS-Plus data to be extracted will not be reported in the main write-up of each behavioural weight management intervention RCT. Hence, it is necessary to extract data from all publications associated with each individual RCT. To complete this, we will adopt a similar approach to literature searching as demonstrated by Orkin et al[23]

The search strategy will be completed in two phases. Phase 1 is identifying 'parent' RCTs. These studies will be identified in two ways. Initially, the behavioural weight loss and behavioural weight loss maintenance interventions included in the USPSTF report will be extracted. Then, we will conduct an update of the literature search used in the USPSTF report to capture recent published trials using the same databases (Medline, CENTRAL and PsychInfo). Databases will be searched from June 2017 (the last date of the USPSTF report) to February 2020. The search strategy includes the following concepts: (1) overweight and obesity AND (2) behavioural weight loss/

behavioural weight loss maintenance interventions. The Medline, CENTRAL and PsychInfo search strategies are outlined in online supplemental file B. In addition to the databases, reference lists of included published primary research and relevant systematic reviews and meta-analyses will be searched for possible further studies for inclusion.

Phase 2 is to 'curate publication families'.[23] This involves identifying publications of any type that relate to the parent RCT through electronic database searching. Authors and study identifiers (such as trial name) will be extracted from each parent RCT, which will then be searched for in the same electronic bibliographical databases as phase 1. Each publication family will be considered as one study.

## Study records
### Data management and study selection
Search results will be imported into EndNote V.X7 bibliographical software, where duplicates will be removed. The literature will then be loaded onto Covidence systematic review software (Veritas Health Innovation, Melbourne, Australia), and title and abstract screening conducted. Piloting of 500 articles will be conducted with minimum two investigators, where differences in interpretation of the inclusion criteria will be discussed between the investigators in order to achieve consistency in the review process. Once this has been completed, the remaining titles and abstracts will be screened for inclusion by minimum two investigators independently. Full-text articles identified as being potentially relevant to the research questions will be accessed and screened by minimum two investigators. Any conflicts will be discussed and resolved by a third reviewer if agreement cannot be reached. For articles excluded at full-text stage, reasons for exclusion will be recorded.

Multiple articles reporting the same study will all be included and amalgamated to ensure all the best available data are used. A PRISMA flow chart will be reported to visualise the study selection.[24]

## Data items
For studies highlighted as eligible for inclusion from the USPSTF report, and those that fulfil the inclusion criteria from our subsequent searches, we will extract data from the reports onto a data extraction form. To ensure that an appropriate breadth and depth of detail is captured, the data extraction form will be based on the Cochrane Public Health Group data extraction form,[25] the Consolidated Standards of Reporting Trials 2010 statement,[26] the Template for Intervention Description and Replication checklist and the PROGRESS-Plus criteria.[4 27]

The following data will be extracted from the studies:
► General information (study authors, publication year, country and source of funding).
► Study information (study aim, design, recruitment location and method, randomisation, blinding and allocation concealment).

► Participant information (measures associated with the PROGRESS-Plus criteria as outlined in table 1).
► Intervention information (content, delivery method, group or individual-level, duration, setting, profession of person delivering intervention).
► Comparator information (control/usual care, content, delivery method, group or individual-level, duration, setting, profession of person delivering intervention).
► Uptake (number of participants invited to trial, number of participants accepting invite, number of participants randomised to intervention arm, number of participants attending >1 session), including impact of PROGRESS-Plus criteria on uptake.
► Adherence/attrition/attendance, including impact of PROGRESS-Plus criteria on these.
► Outcomes (outcomes studies, self-report or objective, follow-up duration, statistical analyses, intervention effect sizes), including impact of PROGRESS-Plus criteria on outcomes.

We will contact authors where data relating to the uptake, adherence and effectiveness outcomes have not been published. The corresponding author for each study will be contacted by email, and followed up after 2 weeks if no response is received. One month from the initial email will be allowed for study authors to respond.

## Risk of bias in individual studies
We will use Cochrane's risk of bias tool for randomised trials (RoB 2) to assess risk of bias across all included studies.[28] This ensures all included studies are assessed by the same criteria for the risk of bias. The tool covers six domains of possible bias: the randomisation process; allocation concealment; participant and trial personnel blinding; blinding of outcome assessment; incomplete outcome data and selective reporting. Each domain is given a ranking of 'low risk', 'high risk' or 'unclear. This will be performed independently by at least two study authors. Where disagreements occur, these will be discussed between authors to reach consensus. A third reviewer will be consulted if agreement cannot be reached. Other possible sources of bias that do not fall within RoB 2's six domains will be noted by reviewers, and commented on if appropriate in the final review. Reviewers will not be blinded to study information (such as study author, institution or journal). Results of the risk of bias assessments will be presented in a summary figure outlining a study's overall risk of bias, as well as the risk of bias in each domain.

## Data synthesis
### Narrative synthesis and Harvest plots
We anticipate that there will be insufficient data to conduct a meta-analysis, therefore, the primary methods of data synthesis will be through narrative analysis and Harvest plots.[29] Harvest plots were proposed by Ogilvie et al as a method for synthesising evidence of the differential effectiveness of population-level public interventions,[29] but

have been used in systematic reviews of various intervention types since.[30–35] Even where there is heterogeneity in measures used, Harvest plots allow for all available and relevant data to be used and presented.[29 36 37] Several study features can be graphically demonstrated on a single plot, such as study quality, statistical significance and sample size. We will present a Harvest plot for each PROGRESS-Plus criteria and whether each trial found a negative, positive or no health inequality gradient; sample size of each study group; and whether the trial considered an intervention or interaction effect on the health inequality gradient.

## Meta-analysis

Should there be sufficient data to conduct a meta-analysis, then the meta-analysis will consider two questions: *are the PROGRESS-Plus criteria associated with the amount of weight loss achieved following behavioural weight management intervention?* and *do the PROGRESS-Plus criteria moderate the effectiveness of behavioural weight management interventions?*. ORs or risk ratios would be pooled for each question; the first question assesses if there is an association between the PROGRESS-Plus criteria and weight loss, the second question considers if there is an interaction. The data would be analysed using Stata V.16 (StataCorp 2019, College Station, Texas, USA), using a random-effects meta-analysis.

Statistical heterogeneity will be assessed using the $I^2$ statistic and its 95% CI. The $I^2$ statistic will be interpreted against the following categorisations: 0%–40% might not be important; 30%–60% may represent moderate heterogeneity; 50%–90% may represent substantial heterogeneity and 75%–100% is likely considerable heterogeneity.[38] The overlap in these categories exists as they are not intended as absolute threshold judgements, but as a guide to be used in conjunction with possible reasons explaining variability.[38] Publication bias will be considered using a funnel plot.

## Patient and public involvement

A patient and public involvement representative reviewed a lay summary of our proposed plan for the systematic review. Feedback was received on the review's aims and definitions of the PROGRESS-Plus criteria. Once the review has been completed, feedback will be sought from the patient and public involvement representatives about the interpretation of findings and plans for an individual participant data meta-analysis.

## ETHICS AND DISSEMINATION

Ethical approval is not required as only aggregate data are going to be acquired and will be used for the purpose for which they were originally collected for. Ethical approval for each trial to be included will have been sought by the original investigators. This systematic review will follow the PRISMA statement.[24]

Inequalities in overweight and obesity, and in health promotion interventions, are widely recognised. However, inequalities in behavioural weight loss interventions delivered or referred to from primary care (or similar) have not yet been considered in a systematic review. This review will identify data on where inequalities in weight loss interventions occur (ie, in which PROGRESS-Plus criteria), and at what stage (uptake, adherence or effectiveness). We anticipate the completed systematic review will be published in a scientific journal, presented at conferences and contribute to the lead author's PhD thesis. The review findings will contribute towards the consideration of intervention-generated inequalities by researchers, policymakers, and healthcare and public health practitioners.

**Acknowledgements** The authors would like to thank Hazel Patel, the patient and public involvement representative, who commented on the lay summary of our proposed plan, for their contribution in the development of this research.

**Contributors** JMB conceived and designed the study, developed the search strategy and drafted the manuscript. SJG conceived the study, contributed to study design and reviewed drafts of the manuscript. MPK contributed to study design and reviewed drafts of the manuscript. ALA conceived the study, contributed to study design and reviewed drafts of the manuscript. All authors have reviewed the manuscript and approved the final version for publication.

**Funding** JMB, SJG and ALA are supported by the Medical Research Council (MRC) (Grant MC_UU_12015/4). The University of Cambridge has received salary support in respect of SJG from the National Health Service in the East of England through the Clinical Academic Reserve.

**Competing interests** ALA is principal investigator on two publicly funded (NIHR, MRC) trials where the intervention is provided by WW (formerly Weight Watchers) at no cost. MPK has undertaken consultancy for Slimming World, and led the clinical and public health guidelines development for NICE from 2005 until 2014.

**Patient consent for publication** Not required.

**Provenance and peer review** Not commissioned; externally peer reviewed.

**ORCID iDs**
Jack Michael Birch http://orcid.org/0000-0001-6292-1647
Amy L Ahern http://orcid.org/0000-0001-5069-4758

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
