## [Reviewer comments · BMJ Open]

ARTICLE DETAILS

TITLE (PROVISIONAL)	Inequalities in the uptake of, adherence to and effectiveness of behavioural weight management interventions: systematic review protocol
AUTHORS	Birch, Jack; Griffin, Simon; Kelly, Michael; Ahern, Amy L.

VERSION 1 – REVIEW

REVIEWER	Dr Dominika Kwasnicka University of Melbourne, Australia
REVIEW RETURNED	19-Aug-2020

GENERAL COMMENTS	Article titled “Inequalities in the uptake of, adherence to and effectiveness of behavioural weight management interventions: systematic review protocol” is a very nicely written protocol for a systematic review that will answer a very topical question regarding inequalities associated with behavioural weight management interventions. Studies show that there is a strong relationship with higher BMI and lower SES. Behavioural interventions that aim to help people lose weight and maintain weight loss long term are common; they often aim to improve population health; however, they may also result in increasing inequalities with mainly affluent people accessing relevant interventions, being able to adhere to them and then to benefit from them. In order to improve population health, we need to understand if health promotion programs are able to address health problems equally for affluent and less affluent individuals. Therefore, this review will answer extremely important question that will help us understand how to best design behavioural intervention to ensure that they are not only effective but that they are also available to all and effective to all individuals in need. This protocol is very nicely written, the introduction sets the scene for the review, and describes relevant literature, the research question and justification for why this review is needed now is well justified. It may be worth adding PROSPERO registration number if the review is already registered now. Strengths and limitations of the review – the authors may consider listing strengths first followed by the limitations to logically match the heading. If it is possible, it could be desirable to avoid some of the acronyms, e.g., IPD, BMW, BMW, USPSTF. It is my personal preference but I am not that keen on acronyms that are not commonly used (BMI, SES, RCT are all okay but other ones really slow down the reader). Study methods are well described, the strategy with updating recent review and looking at different research question is a great idea (and will help to speed up the review process). Table 1 – I believe it's a 'postcode' instead of 'post code'; education – may be difficult to compare between the studies. PROGRESS table – is it
--

	possible to add a note to say where these criteria are taken from? Study eligibility criteria – nicely described. Page 10, line 35, should be ‘RCTs’? Information sources – it seems to me that several authors will need to be contacted and asked for additional data, what will be the strategy to do that? (e.g., two-three email request, maximum one month to get a response). Risk of bias – will the authors assess all included studies or just the ‘new studies’ identified in the current search – could the authors add information if the original review already assessed risk of bias of the included studies? Data synthesis – very nice idea to use Harvest plots. For statistical heterogeneity, I do not understand why % brackets overlap and how to interpret them if they do, is there any scope to explain. Ethics and dissemination plans – all nicely described. It may be worth adding a paragraph on study strengths and limitations again at the end (but I do appreciate that they are highlighted at the start already so up to the authors). I would like to congratulate the authors on such an important piece of research, and I am sure that your review will inform practice and policy.
--	---

REVIEWER	Lucie Nield Sheffield Hallam University
REVIEW RETURNED	12-Oct-2020

GENERAL COMMENTS	A very well-written clear protocol for an interesting study. My only comment is whether you would want to include a measure of behaviour-change treatment integrity (i.e. have the authors reported how they have checked that the behaviour change methodology was delivered consistently and appropriately throughout the intervention) as poor delivery of the intervention may also widen health inequalities.
--

VERSION 1 – AUTHOR RESPONSE

Reviewer: 1

Article titled “Inequalities in the uptake of, adherence to and effectiveness of behavioural weight management interventions: systematic review protocol” is a very nicely written protocol for a systematic review that will answer a very topical question regarding inequalities associated with behavioural weight management interventions. Studies show that there is a strong relationship with higher BMI and lower SES. Behavioural interventions that aim to help people lose weight and maintain weight loss long term are common; they often aim to improve population health; however, they may also result in increasing inequalities with mainly affluent people accessing relevant interventions, being able to adhere to them and then to benefit from them. In order to improve population health, we need to understand if health promotion programs are able to address health problems equally for affluent and less affluent individuals. Therefore, this review will answer extremely important question that will help us understand how to best design behavioural intervention to ensure that they are not only effective but that they are also available to all and effective to all individuals in need.

This protocol is very nicely written, the introduction sets the scene for the review, and describes relevant literature, the research question and justification for why this review is needed now is well justified. It may be worth adding PROSPERO registration number if the review is already registered now.

We have now added the PROSPERO registration number into the abstract and at the beginning of the methods

Strengths and limitations of the review – the authors may consider listing strengths first followed by the limitations to logically match the heading.

The list of strengths and limitations have now been re-ordered to match the heading.

If it is possible, it could be desirable to avoid some of the acronyms, e.g., IPD, BMW, BMW, USPSTF. It is my personal preference but I am not that keen on acronyms that are not commonly used (BMI, SES, RCT are all okay but other ones really slow down the reader).

We have now removed the acronyms that are less commonly used.

Study methods are well described, the strategy with updating recent review and looking at different research question is a great idea (and will help to speed up the review process). Table 1 – I believe it's a 'postcode' instead of 'post code'

The text in Table 1 has been corrected to read 'postcode'

PROGRESS table – is it possible to add a note to say where these criteria are taken from?

Apologies, the was originally omitted in error. The definitions and examples in the table are adapted from a paper by Attwood et al, and the title of the table has been edited to reflect where they are taken from.

Study eligibility criteria – nicely described.

Page 10, line 35, should be 'RCTs'?

This typo has now been corrected

Information sources – it seems to me that several authors will need to be contacted and asked for additional data, what will be the strategy to do that? (e.g., two-three email request, maximum one month to get a response).

The text regarding contacting authors has been moved to the end of the data items section, and now reads:

"We will contact authors where data relating to the uptake, adherence and effectiveness outcomes have not been published. The corresponding author for each study will be contacted by email, and followed up after two weeks if no response is received. One month from the initial email will be allowed for study authors to respond."

Risk of bias – will the authors assess all included studies or just the 'new studies' identified in the current search – could the authors add information if the original review already assessed risk of bias of the included studies?

The original USPSTF review considered risk of bias in terms of an adapted GRADE evaluation of the quality of evidence, hence we felt it useful to use a more explicit risk of bias assessment tool. All included studies will be assessed using the RoB 2, and the text now reflects this more clearly.

"We will use Cochrane's risk of bias tool for randomised trials (RoB 2) to assess risk of bias across all included studies"

Data synthesis – very nice idea to use Harvest plots. For statistical heterogeneity, I do not understand why % brackets overlap and how to interpret them if they do, is there any scope to explain.

The percentage thresholds presented in the protocol paper are taken from Cochrane Consumers and Communication research group. In their guidance, they suggest the categories should be used as rough guidance and not as an absolute judgement, and should be used alongside other information around why there may be high statistical heterogeneity. For clarification, the paragraph now reads:

“Statistical heterogeneity will be assessed using the I² statistic and its 95% confidence interval. The I² statistic will be interpreted against the following categorisations: 0% to 40% might not be important; 30% to 60% may represent moderate heterogeneity; 50% to 90% may represent substantial heterogeneity and; 75% to 100% is likely considerable heterogeneity.[38] The overlap in these categories exist as they are not intended as absolute threshold judgements, but as a guide to be used in conjunction with possible reasons explaining variability [38]. Publication bias will be considered using a funnel plot.”

Ethics and dissemination plans – all nicely described. It may be worth adding a paragraph on study strengths and limitations again at the end (but I do appreciate that they are highlighted at the start already so up to the authors).

As the reviewer notes these are already included previously, and in the interests of word count are not repeated here

I would like to congratulate the authors on such an important piece of research, and I am sure that your review will inform practice and policy.

Reviewer: 2

A very well-written clear protocol for an interesting study. My only comment is whether you would want to include a measure of behaviour-change treatment integrity (i.e. have the authors reported how they have checked that the behaviour change methodology was delivered consistently and appropriately throughout the intervention) as poor delivery of the intervention may also widen health inequalities.

We agree that this could be an interesting avenue to explore, however we believe that this falls outside the scope of this current review.